# Cluster-driven Personalized Federated Recommendation with Interest-aware Graph Convolution Network for Multimedia

Xingyuan Mao*
China University of Petroleum (East China)
Qingdao, China
hsingyuanmao@gmail.com

Yuwen Liu*
China University of Petroleum (East China)
Qingdao, China
yuwenliu97@gmail.com

Lianyong Qi†
China University of Petroleum (East China)
Qingdao, China
Qufu Normal University
Rizhao, China
lianyongqi@upc.edu.cn

Li Duan
Beijing Jiaotong University
Beijing, China
duanli@bjtu.edu.cn

Xiaolong Xu†
Nanjing University of Information Science and Technology
Nanjing, China
xlxu@nuist.edu.cn

Xuyun Zhang
Macquarie University
Sydney, Australia
xuyun.zhang@mq.edu.au

Wanchun Dou
Nanjing University
Nanjing, China
douwc@nju.edu.cn

Amin Beheshti
Macquarie University
Sydney, Australia
amin.beheshti@mq.edu.au

Xiaokang Zhou
Kansai University
Osaka, Japan
zhou@kansai-u.ac.jp

## Abstract

Federated learning addresses privacy concerns in multimedia recommender systems by enabling collaborative model training without exchanging raw data. However, existing federated recommendation models are mainly based on basic backbones like Matrix Factorization (MF), which are inadequate to capture complex implicit interactions between users and multimedia content. Graph Convolutional Networks (GCNs) offer a promising method by utilizing the information from high-order neighbors, but face challenges in federated settings due to problems such as over-smoothing, data heterogeneity, and elevated communication expenses. To resolve these problems, we propose a Cluster-driven Personalized Federated Recommender System with Interest-aware Graph Convolution Network (CPF-GCN) for multimedia recommendation. CPF-GCN comprises a local interest-aware GCN module that optimizes node representations through subgraph-enhanced adaptive graph convolution operations, mitigating the over-smoothing problem by adaptively extracting information from layers and selectively utilizing high-order connectivity based on user interests. Simultaneously, a cluster-driven aggregation approach at the server significantly reduces communication costs by selectively aggregating models from clusters. The aggregation produces a global model and cluster-level models, combining them with the user's local model allows us to tailor the recommendation model for the user, achieving personalized recommendations. Moreover, we propose an adversarial optimization technique to further augment the robustness of CPF-GCN. Experiments on three datasets demonstrate that CPF-GCN significantly outperforms the state-of-the-art models.

## CCS Concepts

• **Information systems → Recommender systems**.

## Keywords

Recommender System, Federated Learning, Graph Convolution Network, Clustering, Multimedia Recommendation

**ACM Reference Format:**
Xingyuan Mao, Yuwen Liu, Lianyong Qi, Li Duan, Xiaolong Xu, Xuyun Zhang, Wanchun Dou, Amin Beheshti, and Xiaokang Zhou. 2024. Cluster-driven Personalized Federated Recommendation with Interest-aware Graph Convolution Network for Multimedia. In *Proceedings of the 32nd ACM International Conference on Multimedia (MM '24), October 28-November 1, 2024, Melbourne, VIC, Australia.* ACM, New York, NY, USA, 9 pages. https://doi.org/10.1145/3664647.3680788

*Xingyuan Mao and Yuwen Liu are co-first authors.
†Corresponding authors.

## 1 Introduction

With the explosive growth of multimedia content in recent years, navigating the vast ocean of available videos, images, and music has become increasingly challenging. Recommender systems have risen to this challenge [12], helping users discover their interested content across various domains [19, 20, 31]. As deep learning technology has become more widely applied [23, 34], it has further enhanced the capabilities of these systems. However, recommender

systems typically process data and train models on central servers, requiring the collection of users' personal data. This exposes users to significant privacy risks. Federated recommender systems have emerged to solve this problem by allowing participants to collaboratively train the model through the exchange of intermediate parameters instead of raw data [17], thereby preserving privacy.

The backbones of existing federated recommender systems are mostly basic, such as Matrix Factorization (MF) [4] and Neural Collaborative Filtering (NCF) [30]. In these methods, updates are primarily according to explicit user-item interactions, overlooking the complex implicit interactions represented by a bipartite graph of users and items. Graph Neural Networks (GNNs) introduce a novel approach to leverage this graph structure [1]. As a prominent GNN variant, Graph Convolutional Networks (GCNs) can effectively gather collaborative information from high-order connections via the propagation of embeddings across the graph [11]. In a federated setting, each client keeps a private graph that is not shared with the server or other users, ensuring privacy protection. Some GCN-based federated recommendation models have been introduced recently [24, 26], where users enhance their local graphs to access high-order neighbor information by uploading encrypted embeddings to the server. However, these frameworks primarily focus on low-order connectivity, limiting their ability to use high-order information as effectively as centralized models. Besides, GCN-based models often encounter the over-smoothing issue, where multiple graph convolution layers cause node embeddings to grow overly similar [18], thereby reducing model effectiveness.

In addition to the issues mentioned above, federated recommender systems also face problems of heterogeneity and high communication costs. Heterogeneity arises because the data owned by various clients is rarely independent and identically distributed (IID) [16], complicating the task of generating only one unified model to effectively serve all users and achieve personalized recommendations, as described in the top left part of Fig. 1. Furthermore, achieving optimal performance in federated recommender systems needs frequent exchanges between the server and all clients. Given that modern recommender systems often rely on complex deep learning architectures with millions of parameters, this results in significant communication burdens.

Motivated by the considerations above, we propose a **C**luster-driven **P**ersonalized **F**ederated Recommender System with Interest-aware **G**raph **C**onvolution **N**etwork (**CPF-GCN**). CPF-GCN comprises two key components: a local interest-aware recommendation module and a cluster-driven global server. For the local client, the interest-aware GCN module divides users into different subgraphs based on their interests, then applies high-order graph convolution operations within these subgraphs. Thus, the learning of a user's embedding can only be facilitated by neighbors who share interests, avoiding the propagation of irrelevant negative noise. For the graph convolution on the subgraphs, we refer to and improve upon the simplified graph convolution proposed in LightGCN [9], introducing an adaptive graph convolution method. This method can adaptively extract information from the ego layer and learn node representations for the next layer, avoiding the inclusion of excessive high-order information and further alleviating the over-smoothing problem. Moreover, we employ an adversarial optimization technique to enhance the robustness of our GCN model.

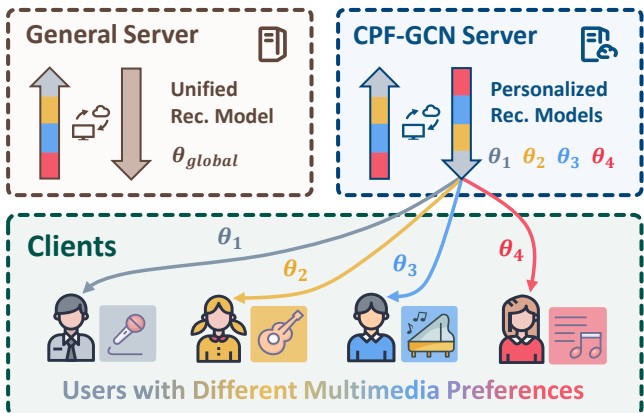

Figure 1: Comparison between a general federated server and our CPF-GCN. Each user uploads distinct preferences to the server. The general server aggregates these preferences and generates a unified recommendation model, while CPF-GCN tailors a personalized model for each user.

For the global server, we propose a cluster-driven federated learning framework. The server first clusters users based on their representations. Unlike traditional federated methods that select all clients or randomly choose some clients for aggregation, our method adaptively selects a certain number of clients from each cluster based on its size, ensuring that the selected clients are representative. In this way, only a relatively small number of clients need to be selected to achieve good performance, reducing communication costs. Subsequently, our model aggregates the chosen clients, generating a global model and several cluster-level models. To safeguard user privacy, a Local Differential Privacy (LDP) technique is used during this procedure. Through the integration of the generated global model, cluster-specific model, and previously obtained local model, a personalized recommendation model can be produced for each user, achieving personalized recommendations while protecting user privacy. As illustrated in the top right part of Fig. 1, users upload their distinct multimedia preferences to the server, then the server tailors a personalized model for each user.

To summarize, our key contributions are outlined as follows:

- We propose a subgraph-enhanced adaptive graph convolution method for the local GCN model to mitigate the issue of over-smoothing by generating subgraphs from the user-item bipartite graph, and adaptively learning node representations by weighing similarities between hidden and ego layers.

- We design a cluster-driven federated framework that selects representatives from each user cluster for model aggregation and generates cluster-level and global models for personalized recommendation, addressing heterogeneity while boosting efficiency.

- We integrate the concept of adversarial training into model optimization by designing an adversarial loss, thereby enhancing the model's robustness.

- Extensive experiments on three multimedia datasets indicate that our proposed CPF-GCN model significantly outperforms existing federated recommendation models.

## 2 Related Work

### 2.1 GNNs for Recommendation

Graph Neural Networks (GNNs) [33] emerge as an effective tool in recommender systems that exploit the interactive information in user-item interaction graphs. Early adaptations of GNNs in recommender systems focused on learning embeddings by capturing the structure of the user-item bipartite graph [6, 15, 21]. For example, [15] designs a removal algorithm to filter redundant information while utilizing GNNs to mine information from high-order connectivity, enhancing the model's generalization capabilities. In addition, models like Graph Convolutional Networks (GCNs) [14] have laid the foundation for subsequent advances in this field. [22] highlights the dynamic nature of both users and points-of-interest, proposing a time-sensitive and interaction-augmented GCN for continuous point-of-interest recommendations. Moreover, over-smoothing is a notorious issue with GNN-based methods, leading to advances that modify or simplify GCN components[9, 18, 27]. LightGCN [9] is an important milestone that simplifies GCNs by eliminating feature transformations and nonlinearities. [18] further proposes IMP-GCN, which addresses the over-smoothing problem by ensuring that high-order graph convolution only involves users with similar interests. [40] designs a layer-refined GCN known as LayerGCN, which reduces the noise introduced by external factors by refining layer representations during each propagation. These works enhance information utilization, dynamic preference learning, and training efficiency. Despite these advancements, privacy protection in recommender systems remains underexplored.

### 2.2 Federated Learning for Recommender Systems

Federated Learning (FL) trains models without exchanging raw data between decentralized devices or servers, keeping user data private. Therefore, FL can be well applied in recommender systems and achieve practical privacy protection [2, 13, 28]. [28] introduces the Federated Averaging algorithm, marking the beginning of its application in various domains, including recommender systems. [2] first combines FL with collaborative filtering for personalized recommendations using implicit user feedback. To address privacy leakage during training and gradient transmission, methods like perturbing data and encrypting gradients are used in some works. [13] randomly selects some unrated items for each user and then assigns some dummy ratings so that the server cannot easily identify the set of scores and rated items during server-client interactions. FedMF [4] uses homomorphic encryption to encrypt item gradients before transferring them to the server. Recently, many advances have begun to combine FL with GNNs to protect users' privacy while capturing higher-order information. FGC [38] is a federated method based on GCNs, using the potential overlap of services across different clients to guide embedding aggregation and sharing, optimizing the local training results. PerFedRec [25] clusters users into different groups, enabling personalized model learning for each cluster. It integrates LightGCN with a FL framework, achieving personalized federated recommendations under privacy protection. However, these methods merely use basic GCN models, which often suffer from the over-smoothing problem, into FL without unique customization for the federated setting.

## 3 Problem Definition

We denote $\mathcal{U} = \{u_1, u_2, \ldots, u_{\mathcal{N}_u}\}$ to be the user set and $I = \{i_1, i_2, \ldots, i_{\mathcal{N}_i}\}$ as the item set. Here, $\mathcal{N}_u$ and $\mathcal{N}_i$ indicate the counts of users and items, accordingly, with the aggregate node count being $\mathcal{N} = \mathcal{N}_u + \mathcal{N}_i$. Consider $\mathbf{A} \in \mathbb{R}^{N \times N}$ as the adjacency matrix representing user-item interactions, where $\mathbf{A}_{n,m} = 1$ if an interaction between user $u_n$ and item $i_m$ is observed. Using this adjacency matrix, we can construct a user-item bipartite graph $G = (\mathcal{V}, \mathcal{E})$, where $\mathcal{V}$ includes user and item nodes, and $\mathcal{E}$ denotes edges. If $\mathbf{A}$ has an entry $\mathbf{A}_{n,m} = 1$, it indicates the presence of an edge connecting user $u_n$ with item $i_m$ within the bipartite graph $G$. Let $G_s$, where $s$ belongs to the set $\{1, \ldots, \mathcal{N}_s\}$, represents a subgraph, with $\mathcal{N}_s$ indicating the total number of subgraphs. The local GCN model takes the aforementioned data as input and iteratively aggregates features from neighboring nodes within subgraphs, facilitating the learning of user and item representations.

In federated settings, historical interactions between users and items are stored on individual user devices rather than centrally on a server. Regarding the graph structure, the user-item bipartite graph $G$ adopts a decentralized approach where every user maintains a private graph, which includes their own interactions with items. However, interactions from a single user alone are insufficient to train the model. To build a graph with multiple user nodes and their interacted items for training the local GCN model, we adopt the strategy employed in [36]. Each user uploads their privacy-preserved embedding along with encrypted item IDs to the server. The server then returns the encrypted user embeddings and item IDs to all users. This allows each user to receive multiple users' embeddings without disclosing identities, enabling them to access neighboring users and expand their local interaction graph $G$.

## 4 Methodology

The framework of the proposed CPF-GCN model is illustrated in Fig. 2. CPF-GCN consists of a local interest-aware recommendation module and a cluster-driven global server. The local GCN model can allocate users into different subgraphs and perform adaptive graph convolution. The server can cluster users and implement aggregation based on these clusters to update the model. They work collaboratively to achieve personalized recommendations.

### 4.1 Local Interest-aware Recommendation Module

For the local client, we propose an interest-aware GCN module that performs subgraph-enhanced adaptive graph convolutions based on user interests. Additionally, we introduce adversarial optimization to further improve the performance and robustness of our module.

*4.1.1 Subgraph Extraction.* This section details the generation of subgraph $G_s$ from the user-item interaction graph $G$, aiming to group users by interests. User grouping operates as an unsupervised classification task, with users being characterized through feature vectors and allocated to clusters. Specifically, each user's feature vector combines information from the graph structure and their ID embedding. We refer to this process as feature fusion:

$$\mathbf{F}_u = \sigma \left( \mathbf{W}_1 \left( \mathbf{e}_u^{(0)} + \mathbf{e}_u^{(1)} \right) + \mathbf{b}_1 \right), \tag{1}$$

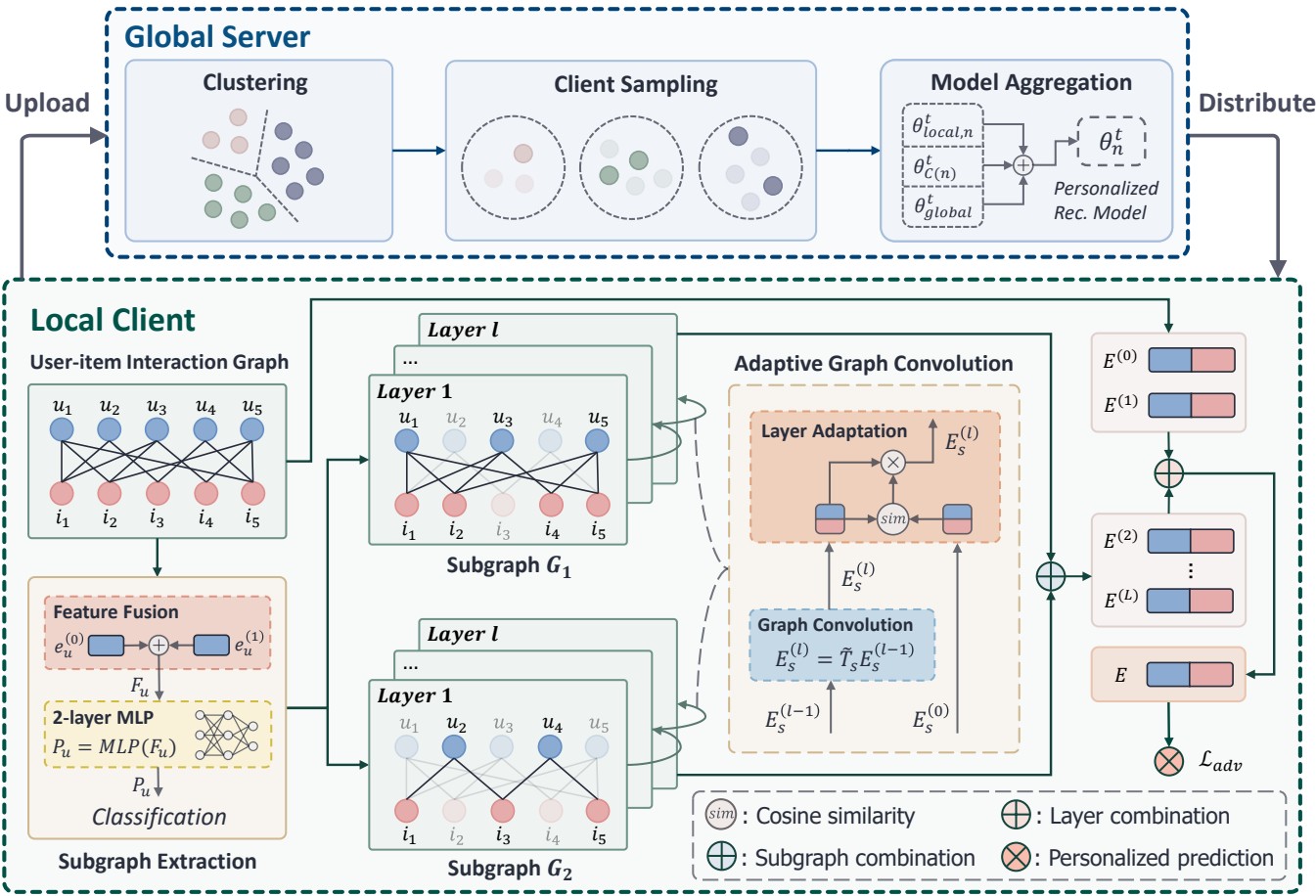

Figure 2: The framework of CPF-GCN. It consists of a local interest-aware GCN module and a cluster-driven global server.

where $\mathbf{F}_u$ represents the user feature obtained through feature fusion. The user embedding of the ego layer is denoted by $\mathbf{e}_u^{(0)}$, and $\mathbf{e}_u^{(1)}$ denotes the user embedding following the first layer of propagation within full graph $G$. The weight matrix and bias vector for the fusion method are respectively symbolized by $W_1 \in \mathbb{R}^{d \times d}$ and $b_1 \in \mathbb{R}^{1 \times d}$, where $d$ indicates the size of the embedding. The activation function $\sigma$ is chosen to be LeakyReLU, given its efficacy in processing either positive or slightly negative information. To assign users to distinct subgraphs, $\mathbf{F}_u$ is further transformed through a two-layer neural network as follows:

$$\begin{aligned}\mathbf{H}_u &= \sigma\left(\mathbf{W}_2\mathbf{F}_u + \mathbf{b}_2\right), \\ \mathbf{P}_u &= \mathbf{W}_3\mathbf{H}_u + \mathbf{b}_3,\end{aligned} \quad (2)$$

where $\mathbf{P}_u$ is the prediction vector indicating the user's group assignment based on the highest value's position within the vector. $\mathbf{W}_2 \in \mathbb{R}^{d \times d}$ and $\mathbf{W}_3 \in \mathbb{R}^{d \times N_s}$ are weight matrices for the two layers of the neural network, while $\mathbf{b}_2 \in \mathbb{R}^{1 \times d}$ and $\mathbf{b}_3 \in \mathbb{R}^{1 \times N_s}$ represent the bias vectors for these layers. Users with similar embeddings will produce similar prediction vectors via Eq. (2), thereby being classified into the same group. The goal of creating subgraphs is to form a matrix that maps user-item relationships within each subgraph. For the matrix of each subgraph, user-item adjacencies from the original graph are excluded if the corresponding user does not belong to the group, based on the group information obtained.

*4.1.2 **Subgraph-enhanced Adaptive Graph Convolution.*** Based on the generated subgraphs, we propose a subgraph-enhanced adaptive graph convolution method for our local GCN model. Since our model is built upon the LightGCN framework, let us first review the graph convolution method in LightGCN. The node embedding of ego layer is represented by $\mathbf{E}^{(0)} \in \mathbb{R}^{N \times d}$. For the $l$-th layer, we represent the node embedding with $\mathbf{E}^{(l)}$. The simplified graph convolution in LightGCN is described by:

$$\mathbf{E}^{(l)} = \tilde{\mathbf{T}}\mathbf{E}^{(l-1)}, \quad (3)$$

where $\tilde{\mathbf{T}}$ is the transition matrix, formulated as $\tilde{\mathbf{T}} = \mathbf{D}^{-1/2}\mathbf{A}\mathbf{D}^{-1/2}$. Here, $\mathbf{A}$ represents the previously defined adjacency matrix, and $\mathbf{D}$ represents the corresponding diagonal matrix where each element $\mathbf{D}_{ii}$ indicates the count of non-zero entries within the $i$-th row of $\mathbf{A}$. This equation signifies that the node embeddings for the $l$-th layer are obtained through the aggregation of data from the $(l-1)$-th layer with the transition matrix $\tilde{\mathbf{T}}$.

However, aggregating features iteratively from higher-order neighbors can cause the over-smoothing problem and a dilution of unique node features. This process overlooks the variety of higher-order features and allows noise from neighbors to degrade the learning of embeddings, consequently diminishing model effectiveness. To address this problem, our proposed model groups users into subgraphs and performs the high-order propagation within these subgraphs. Similarly, the convolution process within a subgraph employs a symmetric transition matrix $\tilde{\mathbf{T}}_s = \mathbf{D}_s^{-1/2}\mathbf{A}_s\mathbf{D}_s^{-1/2}$. The adjacency matrix $\mathbf{A}_s$ for the subgraph $G_s$ is derived from the interaction matrix $\mathbf{R}_s \in \mathbb{R}^{\mathcal{N}_{su} \times \mathcal{N}_{si}}$, with $\mathcal{N}_{su}$ and $\mathcal{N}_{si}$ representing the counts of users and items in the subgraph $G_s$, respectively. Specifically, $\mathbf{A}_s$ is structured as follows:

$$\mathbf{A}_s = \begin{pmatrix} 0 & \mathbf{R}_s \\ \mathbf{R}_s^{\mathrm{T}} & 0 \end{pmatrix}. \tag{4}$$

Given that the immediate connections linking users to items significantly reflect user preferences, in our model, the first-order graph convolution operation incorporates all first-order neighbors. In other words, the first layer embedding propagation acts within the full graph $G$, described by the following formulation:

$$\mathbf{E}^{(1)} = \tilde{\mathbf{T}}\mathbf{E}^{(0)}. \tag{5}$$

For the high-order graph convolution, we perform each convolution operation within subgraphs, and we design an adaptive graph convolution mechanism that adaptively extracts the information from the ego layer to mitigate the introduction of extraneous noise. In this context, a node within a subgraph is restricted to utilizing data solely from its neighbor nodes within the same subgraph. This ensures that users maintain access to information from all interconnected items, as these items are part of the user's subgraph, while keeping the noise from the high-order neighbors out. We formulate the high-order graph convolution within subgraph $G_s$ as:

$$\begin{aligned} \mathbf{E}_s^{(l)} &= \tilde{\mathbf{T}}_s\mathbf{E}_s^{(l-1)}, \\ \mathbf{E}_s^{(l)} &= \left(\mathbf{m}^{(l)} + \xi\right)\mathbf{E}_s^{(l)}, \end{aligned} \tag{6}$$

where $l \geqslant 2$, $\mathbf{E}_s^{(l)}$ represents the embedding of the nodes in subgraph $G_s$ after the $l$-th convolution layer. The similarity vector $\mathbf{m}^{(l)} \in \mathbb{R}^{\mathcal{N}_s}$ captures node similarity between the current layer and the ego layer, aiming to preserve crucial features while adaptively extract information from the ego layer. The small quantity $\xi$ serves to ensure there is no zero vector in $\mathbf{E}_s^{(l)}$. The similarity measurement $\mathbf{m}^{(l)}$ is determined by:

$$\mathbf{m}^{(l)} = sim\left(\mathbf{E}_s^{(l)}, \mathbf{E}_s^{(0)}\right). \tag{7}$$

In this context, cosine similarity serves as the $sim$ function which calculates the similarity between two vectors $\mathbf{e}_i = \mathbf{E}_s^{(l)}[i,:]$ and $\mathbf{e}_j = \mathbf{E}_s^{(0)}[j,:]$ using the formula:

$$sim\left(\mathbf{e}_i, \mathbf{e}_j\right) = \frac{\mathbf{e}_i \cdot \mathbf{e}_j}{\max\left(\|\mathbf{e}_i\|_2\|\mathbf{e}_j\|_2, \epsilon\right)}, \tag{8}$$

where $\epsilon$ represents a tiny constant introduced to prevent division by zero. The adaptation of embeddings enhances the integration of hidden layers akin to the ego layer while diminishing the impact of those layers deviating from it, preserving the uniqueness of each

node's embedding so that they do not become increasingly similar to other nodes. Thus further alleviating the over-smoothing issue.

After getting node embeddings at the $l$-th layer for each subgraph, the ultimate node embeddings for the $l$-th layer are constructed by combining all embeddings for the $l$-th layer from various subgraphs as follows:

$$\mathbf{E}^{(l)} = \sum_{s \in G_s} \mathbf{E}_s^{(l)}. \tag{9}$$

To obtain the final embeddings for users and items, we combine embeddings from all layers, similar to the approach used in LightGCN, which can be expressed by the following equation:

$$\mathbf{E} = \alpha_0\mathbf{E}^{(0)} + \alpha_1\mathbf{E}^{(1)} + \cdots + \alpha_l\mathbf{E}^{(L)}, \tag{10}$$

where $\alpha_l \geq 0$ denotes the weight given to the $l$-th layer. In line with LightGCN, this weight is uniformly set to $\frac{1}{L+1}$, ensuring an equal contribution from each layer to the final embedding.

*4.1.3* ***Personalized Prediction and Adversarial Optimization.*** In our system, each user benefits from a uniquely tailored recommendation model $\theta_n^t$ to achieve personalized multimedia recommendations. Details on formulating $\theta_n^t$ will be provided in Section 4.2.2. Using this personalized model, we can acquire the final representations for users and items. Then, to get personalized predictions, we dot product the final embeddings for each user and item:

$$\hat{y}_{ui} = \mathbf{e}_u\mathbf{e}_i^T, \tag{11}$$

where $\mathbf{e}_u \in \mathbb{R}^d$ and $\mathbf{e}_i \in \mathbb{R}^d$ are the final embeddings of user $u$ and item $i$, respectively.

After getting the prediction, we use an adversarial method to optimize our model. One commonly used loss function in recommender systems is the pairwise Bayesian personalized ranking (BPR) loss [32], designed to promote the model to assign higher scores to observed interactions than those unobserved. The BPR loss can be defined by:

$$\mathcal{L}_{BPR}(\theta) = -\sum_{n=1}^{\mathcal{N}_u} \sum_{i \in \mathcal{I}_u^+} \sum_{j \notin \mathcal{I}_u^+} \ln \sigma\left(\hat{y}_{ui}(\theta) - \hat{y}_{uj}(\theta)\right), \tag{12}$$

where $\theta$ represents the model parameters. $\sigma$ is the sigmoid function, and $\mathcal{I}_u^+$ denotes the items with which $u$ has interacted before.

However, BPR lacks robustness against small data perturbations, leading to potentially poorer generalization on unseen data [10]. To address this issue, we design a loss function that introduces adversarial perturbations to the model parameters during training. We formulate the adversarial loss as follows:

$$\mathcal{L}_{adv}(\theta) = \mathcal{L}_{BPR}(\theta) + \lambda_{adv}\mathcal{L}_{BPR}(\theta + \Delta) + \lambda_{L_2}\|\theta\|^2, \tag{13}$$

where $\lambda_{L_2}$ controls the $L_2$ regularization strength to prevent overfitting and $\lambda_{adv}$ controls the strength of the adversarial component. $\Delta$ denotes the perturbations on model parameters, specifically can be formulated as follows:

$$\Delta = \lambda_\delta \cdot \frac{\nabla_\theta\mathcal{L}_{BPR}(\theta)}{\|\nabla_\theta\mathcal{L}_{BPR}(\theta)\|_2}, \tag{14}$$

where $\lambda_\delta$ is a hyper-parameter controlling the magnitude of the perturbations and $\nabla_\theta\mathcal{L}_{BPR}(\theta)$ represents the gradient of the BPR loss with respect to the model parameters $\theta$. The denominator here

normalizes the gradient to ensure that the perturbation is solely directional, keeping its magnitude controlled by $\epsilon$.

The training process of our proposed adversarial loss can be seen as engaging in a minimax game:

$$\theta^{'}, \Delta^{'} = \arg \min_{\theta} \max_{\Delta} \mathcal{L}_{BPR}(\theta) + \lambda_{adv}\mathcal{L}_{BPR}(\theta + \Delta), \quad (15)$$

where the optimization of model parameters $\theta$ serves as the minimizer, while the perturbations $\Delta$ act as the maximizer, seeking to find the most challenging perturbations against the current model. This game iteratively proceeds until achieving convergence.

By compelling the model to learn more robust and distinctive features, our proposed adversarial optimization method can not only bolster the model's resilience to slight variations in data but also enhance its generalization ability.

## 4.2 Global Cluster-driven Federated Framework

For the global server, we propose a cluster-driven federated learning framework. This framework performs privacy-preserving model and embedding aggregation within different scopes to generate global and cluster-level models, which are used for creating the previously mentioned personalized recommendation model.

### 4.2.1 User Clustering and Sampling.
Similar to the earlier process of extracting local subgraphs, the global server clusters users into different classes based on their embeddings $\mathbf{e}_{u,n}$, with user $n$ being assigned to cluster $C(n)$. Standard clustering techniques like K-means are suitable here. The node representation $\mathbf{e}_{u,n}$ benefits from a combination of characteristic and joint features unique to every user, thereby enriching the representation.

In federated recommender systems, while conventional client selection techniques randomly pick clients from the complete client set, we propose a novel approach that is particularly beneficial under conditions where communication costs are high. Specifically, we introduce a cluster-based client sampling method that adaptively selects a handful of random users from each cluster, relative to the size of the cluster, for participation in the model aggregation. By choosing clients that accurately represent their clusters, our federated model can operate more effectively and efficiently.

### 4.2.2 Server Aggregation and Personalized Model Generation.
Server aggregation incorporates Local Differential Privacy (LDP) as a key privacy-preserving mechanism in our system. LDP, an extension of differential privacy, ensures data confidentiality by encrypting uploaded gradients. This encryption involves clipping gradients whose $L_\infty$-norm exceeds a certain threshold $\delta$ and adding zero-mean Laplacian noise to protect privacy. For each client's gradient $g$, we modify it through the equation:

$$g' = cut(g, \delta) + Lap(0, \lambda), \quad (16)$$

where $g'$ represents encrypted gradient, $cut(\cdot, \delta)$ refers to the cut-off operation at threshold $\delta$, and $Lap(0, \lambda)$ adds Laplacian noise characterized by a scale parameter $\lambda$.

Using the encrypted gradient, our framework executes aggregation for model parameters and embeddings. During aggregation at epoch $t$, the server creates a global model $\theta^t_{global}$ for all users, as well as cluster-level models $\theta^t_{C(n)}$, with $C(n)$ signifying the cluster that includes user $n$. These models are formed using a weighted

**Table 1: Statistics of datasets.**

| Dataset | #Users | #Items | #Interactions | Density |
|---|---|---|---|---|
| Foursquare | 1083 | 38,333 | 91,024 | 0.22% |
| Lastfm-2K | 1,860 | 17,632 | 92,601 | 0.28% |
| MovieLens-100K | 943 | 1,682 | 100,000 | 6.30% |

sum based on local data volumes from participating clients. The update of the global model $\theta^t_{global}$ is given by:

$$\theta^t_{global} = \theta^{t-1}_{global} - \eta \cdot \bar{g}, \quad (17)$$

where $\eta$ represents the learning rate, and $\bar{g}$ is the aggregated gradient for all selected clients $\mathcal{S}$, calculated as a weighted sum of the encrypted gradients $g'_n$:

$$\bar{g} = \sum_{n \in \mathcal{S}} \frac{D_n}{\sum_{i \in \mathcal{S}} D_i} g'_n, \quad (18)$$

where $D_n$ indicates the local data amount of client $n$. Similarly, the update of the cluster-level model $\theta^t_{C(n)}$ are formulated as:

$$\theta^t_{C(n)} = \theta^{t-1}_{C(n)} - \eta \cdot \bar{g}_{C(n)}, \quad (19)$$

where $\bar{g}_{C(n)}$ is the aggregated gradient for the cluster $C(n)$, calculated as a weighted sum of the encrypted gradients $g'_n$ from the selected clients within the cluster.

In the following step, we tailor a personalized recommendation model for each user as described in Section 4.1.3. Consider $\theta^t_{local,n}$ as the local GCN model learned at epoch $t$ for user $n$. The global model $\theta^t_{global}$, and the cluster-level model $\theta^t_{C(n)}$, combined with $\theta^t_{local,n}$, are used to generate a personalized recommendation model $\theta^t_n$ for user $n$ using the formula:

$$\theta^t_n = \beta_{n,1}\theta^t_{local,n} + \beta_{n,2}\theta^t_{C(n)} + \beta_{n,3}\theta^t_{global}, \quad (20)$$

where $\beta_{n,1}, \beta_{n,2}, \beta_{n,3}$ are the weights that balance the contributions of the local, cluster-level, and global models, respectively. These coefficients can be either fixed or learnable parameters.

## 5 Experiment

### 5.1 Experimental Setup

#### 5.1.1 Datasets.
We use three datasets enriched with multimedia information, including **Foursquare** [37], a dataset capturing user check-ins; **Lastfm-2K** [3], a collection of music listening histories from the Last.fm platform; and **MovieLens-100K** [8], a dataset of movie ratings by users. All these datasets are publicly available and commonly used to evaluate multimedia recommender system models, the key characteristics of which are shown in Table 1.

#### 5.1.2 Baseline Methods.
We conduct comparisons between our CPF-GCN and several representative or cutting-edge models to prove its efficacy. These methods include centralized approaches as well as federated methods, specifically as follows:

The centralized models:

- **BPR [32]:** It is a typical matrix factorization algorithm optimized through Bayesian personalized ranking (BPR) loss.

**Table 2: Overall performance comparison. The best results are in bold and the second-best results are underlined.**

| Dataset | Metric | Centralized Baselines | | | Federated Baselines | | | | | | CPF-GCN | Improv. |
| | | BPR | NGCF | LightGCN | FedMF | FedNCF | FedGNN | PerFedRec | FedIS | PFedRec | | |
|---|---|---|---|---|---|---|---|---|---|---|---|---|
| Foursquare | Recall@5 | 0.3176 | 0.3490 | 0.3601 | 0.3324 | 0.3209 | 0.3361 | 0.3878 | 0.3223 | 0.4033 | **0.4515** | +12.0% |
| | NDCG@5 | 0.2307 | 0.2599 | 0.2728 | 0.2298 | 0.2154 | 0.2399 | 0.3144 | 0.2301 | 0.2913 | **0.3768** | +19.8% |
| | Recall@10 | 0.4044 | 0.4404 | 0.4552 | 0.4304 | 0.4203 | 0.4146 | 0.4552 | 0.4312 | 0.4782 | **0.5171** | +8.1% |
| | NDCG@10 | 0.2589 | 0.2909 | 0.3023 | 0.2778 | 0.2627 | 0.2690 | 0.3361 | 0.2638 | 0.3142 | **0.3965** | +18.0% |
| Lastfm | Recall@5 | 0.7014 | 0.7043 | 0.7269 | 0.6930 | 0.6676 | 0.6817 | 0.6258 | 0.7258 | 0.7426 | **0.7624** | +2.7% |
| | NDCG@5 | 0.5956 | 0.6183 | 0.6062 | 0.5653 | 0.5341 | 0.5496 | 0.5523 | 0.6109 | 0.6184 | **0.6814** | +10.2% |
| | Recall@10 | 0.7405 | 0.7822 | 0.7892 | 0.7643 | 0.7312 | 0.7629 | 0.6973 | 0.7833 | 0.7862 | **0.8016** | +1.6% |
| | NDCG@10 | 0.6023 | 0.6301 | 0.6272 | 0.5820 | 0.5598 | 0.5750 | 0.5773 | 0.6313 | 0.6440 | **0.6932** | +7.6% |
| MovieLens | Recall@5 | 0.4563 | 0.4624 | 0.4687 | 0.4782 | 0.4379 | 0.3606 | 0.5736 | 0.5901 | 0.5776 | **0.6829** | +15.7% |
| | NDCG@5 | 0.3231 | 0.3243 | 0.3209 | 0.3427 | 0.2930 | 0.2437 | 0.4580 | 0.4502 | 0.4277 | **0.5649** | +23.3% |
| | Recall@10 | 0.6238 | 0.6257 | 0.6320 | 0.6489 | 0.6115 | 0.5345 | 0.6607 | 0.7205 | 0.7189 | **0.7709** | +7.0% |
| | NDCG@10 | 0.3563 | 0.3755 | 0.3721 | 0.3902 | 0.3417 | 0.2966 | 0.4736 | 0.5052 | 0.4780 | **0.5982** | +18.4% |

- **NGCF [35]:** It is a representative GNN-based model that propagates embeddings inside a bipartite graph to capture collaborative signals through high-order connectivities.
- **LightGCN [9]:** This model simplifies NGCF by eliminating the feature transformation and nonlinearities. It implements a light graph convolution method for message propagation.

The federated models:

- **FedMF [4]:** This model uses federated matrix factorization and employs stochastic gradient descent while safeguarding gradients through a secure aggregation method.
- **FedNCF [30]:** This is a federated recommendation model built upon the neural collaborative filtering (NCF) framework. It can train NCF models while preserving user privacy.
- **FedGNN [36]:** It utilizes the GNN for privacy-preserving federated recommendation, using LightGCN as the backbone model for local clients.
- **PerFedRec [25]:** It is a federated recommendation model enhanced with a joint representation learning technique to achieve personalized recommendations.
- **FedIS [5]:** It is an efficient federated recommendation model that can reduce user feature dependencies through the training of a global item-based collaborative filtering model.
- **PFedRec [39]:** It proposes a dual personalization approach to achieve detailed customization for both users and items, allowing slight adjustments to item embeddings.

*5.1.3 **Evaluation Metrics**.* We apply the commonly used leave-one-out method for evaluation [35]. We sort the users' behaviors by time. For each user in the dataset, their most recent interaction is designated for testing, while all prior interactions serve as training data. Following previous settings [25, 29], we select 100 uninteracted items randomly and place these items in the test set, then rank the test item within this set. The evaluation protocols we used are Recall@$k$ and NDCG@$k$, where $k$ is chosen to be 5 or 10, using the top-5 and top-10 results to compute the performance.

*5.1.4 **Hyper-parameter Settings**.* Our CPF-GCN model is implemented in PyTorch. The number of training epochs is set to 500 with an early stopping strategy. The embedding size $d$ is fixed to 64

**Table 3: Ablation study on three datasets. R@5 and N@5 refer to Recall@5 and NDCG@5, respectively.**

| Model | Foursquare | | Lastfm | | MovieLens | |
| | R@5 | N@5 | R@5 | N@5 | R@5 | N@5 |
|---|---|---|---|---|---|---|
| CPF-GCN-$s$ | 0.4174 | 0.3530 | 0.7091 | 0.6370 | 0.5546 | 0.4349 |
| CPF-GCN-$p$ | 0.3259 | 0.2423 | 0.6694 | 0.5480 | 0.4274 | 0.2889 |
| CPF-GCN-$a$ | 0.4395 | 0.3751 | 0.7543 | 0.6756 | 0.6702 | 0.5508 |
| CPF-GCN | **0.4515** | **0.3768** | **0.7624** | **0.6814** | **0.6829** | **0.5649** |

for both users and items, and the embeddings are initialized using the Xavier method [7]. The learning rate is initially configured at 0.01, and we apply a dropout ratio of 0.3. For the adversarial settings, the coefficients $\lambda_{adv}$ and $\epsilon$ are both set to 0.1 to control the strength of adversarial perturbations. There are 128 users that take part in each training round, and we defaultly divide these users into 3 clusters. The parameters $\beta_{n,1}, \beta_{n,2}, \beta_{n,3}$, which are used in generating personalized recommendation model, are all fixed to $\frac{1}{3}$.

## 5.2 Performance Comparison

The performance comparison outcomes are shown in Table 2. The results show that compared with the baseline models, our proposed model consistently provides better performance. The most significant improvement is seen in the MovieLens dataset, since each user in this dataset has interacted with a large number of items, which helps us tailor personalized recommendation models for them. Specifically, we can find that CPF-GCN achieves a significant lead over other models in terms of NDCG, indicating that our personalized model excels in capturing relevant items and optimizing their ranking order. Although the lead in recall is not as high as that in NDCG, there is still a noticeable improvement. Additionally, for different top-$k$ values, our model shows a more significant lead at $k = 5$ compared to $k = 10$. This suggests that our model is particularly effective at ranking the most relevant items at the top of the recommendation list, allowing us to recommend the multimedia content that users are most interested in.

## 5.3 Ablation Study

To prove the impact of some important modules of CPF-GCN, we perform ablation experiments on three datasets. We implement

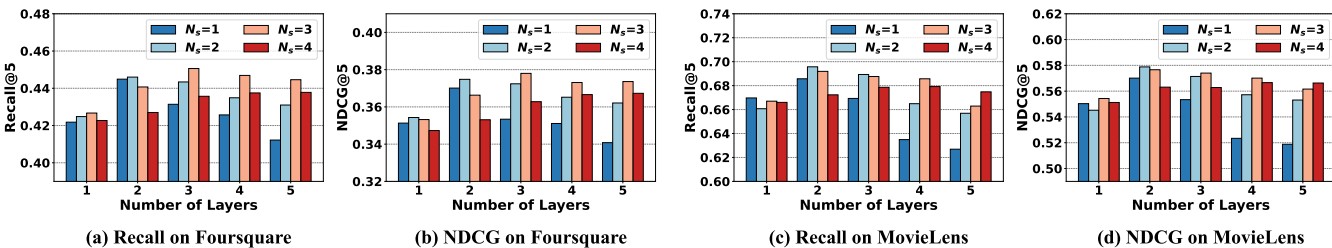

Figure 3: Effect of number of layers and subgraphs.

three variants of our model: **CPF-GCN-*s*** replaces the subgraph-enhanced adaptive graph convolution method with the primary simplified graph convolution introduced in LightGCN. **CPF-GCN-*p*** modifies CPF-GCN by removing the user clustering step and personalized recommendation module, thereby creating a unified model for all users. **CPF-GCN-*a*** removes the adversarial optimization method, using the BPR loos function instead. Table 3 displays the outcomes of ablation experiments in terms of Recall@5 and NDCG@5. We can find that the full CPF-GCN model outperforms all its variants. Specifically, the **CPF-GCN-*p*** model shows the poorest performance, proving that our proposed methods can effectively customize personalized recommendation models for users, achieving more accurate recommendations. The performance also declines in **CPF-GCN-*s***, demonstrating the superiority of our proposed local interest-aware GCN model over existing GCNs. The performance of **CPF-GCN-*a*** also decreases compared to the full CPF-GCN, indicating that adversarial optimization contributes to the improvement of model robustness and generalization performance.

## 5.4 Hyper-parameter Analyses

In this section, we explore how some key hyper-parameters affect the performance of the CPF-GCN. These parameters include the number of layers and subgraphs, the coefficient $\lambda_{adv}$, which controls the strength of the adversarial loss, and the number of user clusters.

*5.4.1 **Effect of Number of Layers $L$ and Subgraphs $\mathcal{N}_s$.*** We implement CPF-GCN with varying numbers of layers $L$ and subgraphs $\mathcal{N}_s$, as shown in Fig. 3. From the perspective of $L$, CPF-GCN performs best with $L = 2$ or $L = 3$ on two datasets. As we stack more than three layers, the performance of models with $\mathcal{N}_s = 1$, i.e., only one graph, significantly declines, indicating the issue of over-smoothing. In contrast, models with multiple subgraphs show more stable performance, suggesting our method can effectively mitigate the over-smoothing problem. Regarding $\mathcal{N}_s$, models with $\mathcal{N}_s = 2$ and $\mathcal{N}_s = 3$ perform better, while the $\mathcal{N}_s = 4$ model, despite being the most stable with increased layers, also cuts off many important connections between short-distance nodes. As a result, its performance after stacking more layers is even worse than that of models with fewer layers and subgraphs, and additionally leads to high communication costs due to the larger number of subgraphs.

*5.4.2 **Effect of Adversarial Coefficient $\lambda_{adv}$.*** $\lambda_{adv}$ is a coefficient that controls the strength of the adversarial loss. The performance of CPF-GCN with different $\lambda_{adv}$ is shown in Fig. 4. From the results, we can see that the two datasets have a significant difference in sensitivity to $\lambda_{adv}$. For the Foursquare dataset, CPF-GCN performs best when $\lambda_{adv} = 10$. In contrast, for the MovieLens dataset,

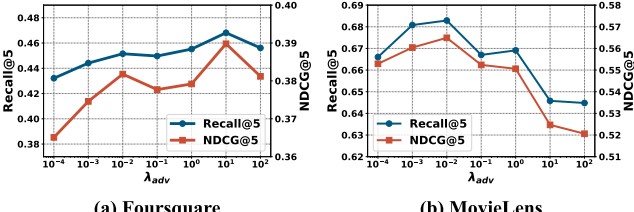

(a) Foursquare         (b) MovieLens

Figure 4: Effect of $\lambda_{adv}$.

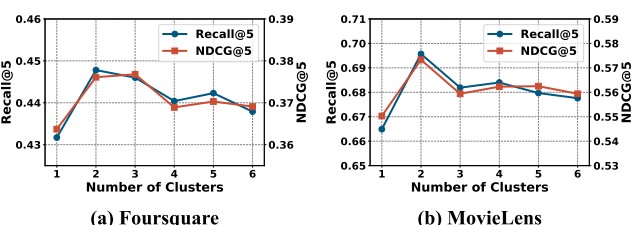

(a) Foursquare         (b) MovieLens

Figure 5: Effect of number of user clusters.

the best performance is observed at $\lambda_{adv} = 0.01$, after which the model performance significantly decreases. This indicates that excessive artificial perturbations may disrupt the training process and reduce the model performance, requiring us to identify the most appropriate value of $\lambda_{adv}$ for different datasets.

*5.4.3 **Effect of Number of User Clusters $K$.*** User clusters are generated in the global server to achieve personalized recommendations. Fig. 5 illustrates the impact of different numbers of user clusters $K$. The results show that compared to $K = 1$, where all users are in the same cluster, dividing users into two or more clusters enhances the model's recommendation performance. CPF-GCN performs best when $K = 2$, and although the performance slightly decreases as $K$ increases, it still remains better than when $K = 1$.

## 6 Conclusion

In this paper, we introduce a GCN-based personalized federated multimedia recommendation framework named CPF-GCN, which addresses the challenges of over-smoothing, data heterogeneity, and high communication costs associated with existing GCN-based federated recommender systems. By integrating the local interest-aware GCN module with the global cluster-driven federated framework, CPF-GCN can effectively capture user preference, achieving personalized multimedia recommendations while preserving user privacy. Extensive experiments on three multimedia datasets demonstrate that CPF-GCN significantly outperforms state-of-the-art federated recommendation models.

## Acknowledgments

This work was supported by the National Natural Science Foundation of China (No. 92267104, No. 62372242, No. 62072159), Natural Science Foundation of Shandong Province (No. ZR2023MF007), Beijing Nova Program (No. 20230484257), Grants-in-Aid for Scientific Research (C) from Japan Society for the Promotion of Science (JSPS) under Grant 23K11064. Dr. Xuyun Zhang is the recipient of an ARC DECRA (No. DE210101458) funded by the Australian Government.

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
