# OpenReview forum: "Cluster-driven Personalized Federated Recommendation with Interest-aware Graph Convolution Network for Multimedia"
_acmmm.org/ACMMM/2024/Conference — MM2024 Poster_

### Official Review · Reviewer_PYBV · 2024-05-14

**Rating:** 3
**Confidence:** 3

**Summary:**

Graph Convolutional Networks (GCNs) face challenges in federated settings due to problems such as over-smoothing, data heterogeneity, and elevated communication expenses. This paper proposes a Cluster-driven Personalized Federated Recommender System with Interest-aware Graph Convolution Network (CPF-GCN) for multimedia recommendation. CPF-GCN comprises a local interest-aware GCN module that optimizes node representations through subgraph-enhanced adaptive graph convolution operations, mitigating the over-smoothing problem by adaptively extracting information from layers and selectively utilizing high order connectivity based on user interests. Simultaneously, a cluster-driven aggregation approach at the server achieves personalized recommendations by selectively aggregating models from clusters. Experiments on three multimedia datasets demonstrate that CPF-GCN significantly outperforms the state-of-the-art models.

**Strengths:**

(1) Federated recommendation represents a promising research direction. Its design, particularly in addressing issues of heterogeneity and high communication costs, stands out as the crucial challenges to tackle.

(2) This paper introduces an innovative approach to address the challenge of over-smoothing in local Graph Convolutional Network (GCN) models, a crucial issue in the field of graph learning.

(3) The experiments of this paper seem to be very solid and convincing.

**Limitations:**

(1) Using a cluster-driven aggregation approach to achieve personalization is not so novel. It is essential to note that many previous researches have explored clustering methods for personalized federated learning. Regrettably, the paper lacks thorough elucidation regarding the distinctions between the clustering method proposed herein and those from prior works, as well as fails to highlight the unique advantages of clustering offered by this paper.

(2) Some parts of the paper are pretty vague. For example, (1) Section 3 is not doing a good job of explaining federated recommendations. The relationship between the clients in a FL system and the user nodes in the graph remains ambiguous. The paper appears to suggest the existence of a graph $G$ on each client, housing multiple user nodes. However, this premise contradicts real-world scenarios. For example, a user's phone (i.e. client) typically only accesses one user's account and cannot aggregate information from other users. (2) The paper lacks clarity regarding the combination of embeddings between subgraphs when different subgraphs contain distinct users and items in Section 4.1.2.

(3) The paper would benefit from elucidating the intrinsic connection and components interaction between Subgraph Extraction, Personalized Recommendation, and Advisory Optimization. At present, these three components appear as a stack of seemingly unrelated technologies.

(4) The adversarial optimization method proposed in Section 4.1.4 bears striking similarities to the technology outlined in "BPR: Bayesian personalized ranking from implicit feedback." It would be beneficial for you to elucidate the distinct innovative features of your method.

**Suitability:**

3

---

### Official Review · Reviewer_f6KA · 2024-05-24

**Rating:** 5
**Confidence:** 3

**Summary:**

The paper introduces CPF-GCN, a federated learning framework for multimedia recommendation that addresses privacy concerns by training models collaboratively without sharing raw data. CPF-GCN utilizes an interest-aware graph convolutional network (GCN) module to capture complex user-item interactions and reduce over-smoothing by adaptively extracting information from user subgraphs based on interests. Additionally, it employs a cluster-driven aggregation approach on the server to decrease communication costs and generate personalized recommendations by combining global, cluster-level, and local user models. Adversarial optimization further enhances the robustness of the model. Experiments on three multimedia datasets show CPF-GCN significantly outperforms state-of-the-art federated recommendation models, achieving personalized and privacy-preserving recommendations.

**Strengths:**

1. The motivations are clear. This work aims to address the issues, such as over-smoothing (the perspective of GNN), data heterogeneity (the perspective of FL), and elevated communication expenses (the perspective of FL)
2. The proposed framework is comprehensive. Several subcomponents are designed to address these issues.
3. The presentation is good. The organization facilitates readability. Figures 1,2 describe the settings and proposed frameworks.

**Limitations:**

1. The issues that motivate this work are general in the field of GNN or FL. In the multimedia settings, what more the particular or new aspects of these issues?
2. Some details should be clarified. In Section 4.1.1 Subgraph Extraction, how to optimize this unsupervised classification task? Please provide the loss function or optimization objective.

**Suitability:**

2

---

### Official Review · Reviewer_8aJu · 2024-05-31

**Rating:** 4
**Confidence:** 3

**Summary:**

The paper introduces a novel multimedia recommender system called Cluster-driven Personalized Federated Recommender System with Interest-aware Graph Convolution Network (CPF-GCN), which leverages federated learning to address privacy concerns while enhancing recommendation accuracy. By utilizing Graph Convolutional Networks (GCNs), CPF-GCN aims to capture complex implicit interactions between users and multimedia content without the need to share raw data. Traditional models like Matrix Factorization (MF) are limited in their ability to model these interactions, which CPF-GCN overcomes by employing a local interest-aware GCN module. This module optimizes node representations through subgraph-enhanced adaptive graph convolution operations, effectively mitigating issues such as over-smoothing by selectively utilizing high-order connectivity based on user interests.

**Strengths:**

The integration of GCNs allows the system to exploit high-order neighbor information, providing a deeper understanding of user-content interactions.

The model adapts to individual user preferences by incorporating local user data into the global learning process, leading to more accurate and personalized recommendations.

**Limitations:**

The experiments are conducted on a relatively small dataset, which may not convincingly demonstrate the model's effectiveness across diverse and larger multimedia environments.

Despite focusing on multimedia recommendation, the model does not explicitly incorporate multimedia-specific information, which could limit its ability to fully understand and utilize the content features.

**Suitability:**

2

---

### Meta-Review · Area_Chair_9Xnh · 2024-07-05

**Recommendation:** Accept (Poster)
**Confidence:** 5

**Metareview:**

The motivation is clear, and a comprehensive and innovative framework is developed with subcomponents (subgraph extraction, personalized recommendation, and advisory optimization) designed to address the issues. High-order information exploitation by integrating GCN architecture also provides insights of user-content interactions. In general, this paper is easy to follow and interesting.